# Effect of Process Parameters on Electrodeposition Process of Co-Mo Alloy Coatings

Xiang Nan [1] , Fu Wang [1], Sensen Xin [2], Xuewei Zhu [1],*  and Qiongyu Zhou [3],*

1   College of Mechanical and Electronic Engineering, Northwest A&F University, Xianyang 712100, China
2   Guangdong Midea Group Kitchen & Bathroom Appliance Manufacturing Co., Ltd., Foshan 528311, China
3   School of Materials Science and Hydrogen Energy, Foshan University, Foshan 528000, China
*   Correspondence: author: zxw_83614@163.com (X.Z.); zhouzhouqiongyuxf@126.com (Q.Z.);
    Tel.: +86-029-15209261723 (X.Z.)

**Abstract:** Plating bath composition, temperature, current density, and pH have a great influence on the properties of a Co-Mo alloy coating. However, these conclusions are obtained from the perspective of test results. Most of the factors that affect the properties of the coating operate by interfering with the electrodeposition process. Therefore, it is of great significance to study the kinetics of the electrodeposition process. To further study the influence of process parameters on the deposition process of Co-Mo alloy coatings, cyclic voltammetry (CV), scanning linear voltammetry (LSV), and electrochemical impedance testing (EIS) were used to study the deposition kinetics of Co-Mo alloy coatings. The results show that with the increase in sodium molybdate content in the plating solution, more Mo elements are involved in the deposition process. At the same time, a higher Mo element content can reduce the dissolution of the coating as the anode and improve the service life of the coating, but too high a Mo element content will lead to cracks on the surface of the coating. The deposition temperature will affect the processes of reduction deposition and oxidation dissolution of the coating. The pH of the plating bath will directly affect the reduction reaction process of $MoO_4^{2-}$ ions in the plating bath. With the increase in the pH value, the reduction reaction rate of $MoO_4^{2-}$ ions decreases, and the cathodic reduction reaction current density decreases. At the same time, the peak current density of anodizing decreases with the decrease in the pH value of the plating solution, indicating that the alkaline plating solution has an inhibitory effect on the plating dissolution process.

**Keywords:** Co-Mo alloy coating; process parameters; electrodeposition process; kinetics





## 1. Introduction

Chromium plating is one of the most common protective coatings on metal surfaces and has many advantages, such as excellent corrosion resistance, high hardness, high wear resistance, and an aesthetic appearance [1–3]. However, hexavalent chromium ions in the chromate plating solution used in the traditional chromium plating process have strong carcinogenicity, which is not conducive to environmental protection and health [3–6]. Although alternative schemes of trivalent chromium plating have been proposed [7–9], the reliability of trivalent chromium plating is not good enough, and the micro-discontinuous surface of the plating develops, forming nodular and fragmentary deposits and micro-cracks, so it is not widely used [10–12]. With the strengthening of people's awareness of environmental protection and the strengthening of the supervision of environmental protection departments, it is becoming increasingly urgent to develop clean and environmentally friendly chromium replacement coating materials. Mo and Cr are located in the same main group with similar properties, and Mo has a favorable influence on the performance of the metal coating, improving its hardness, wear resistance, and corrosion resistance [13–16], and can be used as an alternative Cr element in the coating.

An individual Mo element cannot be effectively deposited, but it can be deposited with the help of the induction of Fe group elements (Fe, Co and Ni) and other elements [17,18].

Therefore, common Mo alloy coatings often exist in the form of Co-Mo [19–22], Ni-Mo [23–25], and other alloys. In Fe group elements, the Co atom is a densely packed hexagonal structure with relatively high mechanical properties such as hardness. At the same time, during the deposition process, Co will produce a Co-rich transition layer on the substrate surface, which is more conducive to improving the performance of the coating.

Stefania Costovici et al. [20] proposed several electrodeposition processes for Ni-Mo and Co-Mo alloy coatings, including deep eutectic solvents (DES). In addition, a new electrolyte system based on a choline chlorine–urea–citric acid mixture (ILC) has been developed. Eva Pellicer et al. [19] found that Co-Mo alloys with high Mo content (35–40 wt.%) would produce many cracks in the process of DC electrodeposition and that the reverse pulse plating method reduced the stress of the Co-Mo alloy and improved its mechanical properties. H. Krawiec et al. [21,22] added $TiO_2$ nanoparticles to the Co-Mo alloy coating and prepared a Co-Mo/$TiO_2$ nanocomposite coating by electrodeposition. They found that the Co-Mo/$TiO_2$ nanocomposite coating is tougher than pure Co, with low residual macro-stress and very good wear resistance. In addition, it has excellent corrosion resistance. However, in these research processes, the study of technological parameters involves only a simple description and is not involved in the study of electrodeposition kinetics.

However, during the study of alloy coatings, most of the factors that affect the properties of the coating operate by interfering with the electrodeposition process. Therefore, it is of great significance to study the kinetics of the electrodeposition process. In this context, a series of studies on the deposition kinetics of a Co-Mo alloy coating with high corrosion resistance on the surface of low-carbon steel, which is widely used in mechanical materials, are presented in this paper.

## 2. Materials and Methods

### 2.1. Sample Preparation

In this study, $10 \times 10$ mm (exposed area 1 $cm^2$) Q235 steel (Table 1) was selected as the base material. Before deposition, samples were welded with wire, ground with sandpaper (400–2000 mesh), cleaned, degreased with acetone and flushed, acidified with 10% HCl and flushed, washed with alcohol, and dried. All samples in this paper were prepared by direct current electrodeposition in a plating bath containing $CoSO_4 \cdot 7H_2O$ (0.14 mol/L), $Na_2MoO_4 \cdot 2H_2O$ (0.0047–0.0233 mol/L), and $Na_3C_6H_5O_7 \cdot 2H_2O$ (0.2 mol/L). According to Elvira Gómez et al. [26], in the alkaline plating solution, $Na_3C_6H_5O$ is the complexing agent and plays an important role in the plating solution. The addition of $Na_3C_6H_5O$ allows $Co^{2+}$ in the coating to exist in the form of the citrate complex $CoCit^-$ and reduce the $Co^{2+}$ loss. At the same time, $CoCit^-$ is also an important part of $MoO_4^{2-}$ that is induced to change into the metal Mo during the deposition process. In addition, sodium citrate and the $H^+$ dissociation equilibrium can play a buffering role. According to their results, 0.2 mol/L citrate assured coherent and homogeneous Co-Mo deposits. The electrochemical workstation was used as the power source. The coating was prepared with a three-electrode system, that is, the working electrode (WE) was connected with a platinum plate as the positive electrode, and the reference electrode (RE) and the counter electrode (CE) were connected with a substrate as the negative electrode. We adjusted the plating solution pH (6–11) by adding NaOH and $H_2SO_4$, and used a constant temperature magnetic agitator to stir the composite plating solution (280–300 rad/min) before and during plating. We selected a current density of 30 mA·cm$^{-2}$, a temperature of 30–70 °C, and a plating solution that was transparent without internal insoluble matter. After this, the treated substrate was placed into the plating solution for deposition, which was deposited in 3600 s, and the Co-Mo coating was prepared by adjusting the technological parameters. When placing the substrate sample into the solution, the distance between the substrate and the platinum electrode should be adjusted in advance, and the substrate and the platinum electrode should be placed "face to face" as far away as possible to ensure the uniform distribution of the current on the substrate surface during deposition.

**Table 1.** Chemical composition of the Q235 steel (wt.%).

| Fe | C | Mn | Si | S | P |
|---|---|---|---|---|---|
| Balance | 0.142 | 0.311 | 0.330 | 0.041 | 0.043 |

*2.2. Coating Characteristics*

The surface morphology was studied using secondary electron imaging with a scanning electron microscope (HITACHIS-3400 scanning electron microscope of Hitachi, Hitachi High-tech Company, Tokyo, Japan). According to the morphology of the coating, a magnification of 1000 to 3000 times was selected. The element distribution and chemical composition of the coating were determined by an energy dispersive spectrometer (EDS, Oxford Instruments, Oxford, UK). An X-ray diffractometer (a SmartlabSE-type X-ray diffractometer produced by Nishiko Co., Ltd., Osaka, Japan) was used. The Cu target was selected as the experimental target, and the incident wavelength $\lambda = 1.54$ A (0.154 nm) was used for X-ray diffraction to obtain the microstructure of the coating.

The Parstatversat3F electrochemical station was used to measure the corrosion electrochemical performance of the coating sample. A three-electrode system was used for wiring, that is, the prepared coating sample was used as the working electrode (WE), the saturated calomel electrode (SCE, Ametek Trading Co., LTD, Shanghai, China) as the reference electrode (RE, Ametek Trading Co., LTD, Shanghai, China), and the Pt electrode (20 mm × 20 mm) as the auxiliary electrode (CE, Ametek Trading Co., LTD, Shanghai, China). A 3.5 wt.% NaCl solution (Xilong Scientific Co., Ltd., Guangdong, China) was used as the corrosion medium, and the corrosion resistance of the coating was tested in this solution. All electrochemical tests were carried out at room temperature if there was no operating temperature limitation.

The main purpose of the corrosion open-circuit potential test (OCP) is to track how the electrode potential of the sample changes over time after immersion in corrosive solution without loading. The coating sample was exposed to 3.5 wt.% NaCl solution as the working electrode, and the scanning speed was set at 0.2 mV/s. After the soaking time reached 45–55 min, the open-circuit potential reached a stable state.

The transformation relationship between the potential and the current density at the electrode can be described by the polarization curve (LSV). The cathodic polarization curve is used in this paper to study the kinetics of coating deposition. The sample to be tested was immersed in a 3.5 wt.% NaCl solution to measure the open-circuit potential for 1 h. After the open circuit potential was stabilized, negative scanning started from ($E_{ocp}$ + 0.0 V) to ($E_{SCE}$ −1.5 V), and the scanning speed was set at 1 mV/s. At the same time, the restriction condition was applied, that is, the reaction current density of the electrode surface did not exceed 1 mA/cm$^2$. After the test was completed, the sample was taken out, rinsed with deionized water, and air-dried.

Cyclic voltammetry (CV) can obtain the kinetic process and reaction mechanism and change the characteristics of the electrode reaction process. These reaction processes are obtained by measuring the change in the current on the electrode surface under the voltage cycle change. In this paper, cyclic voltammetry was used to analyze the deposition kinetics of the coating under different process parameters. When the potential stabilized, the negative scanning started from ($E_{SCE}$ −0.1 V) to ($E_{SCE}$ −1.25 V), and when the potential reached ($E_{SCE}$ −1.25 V), the positive scanning started at ($E_{SCE}$ −0.1 V), and then the negative scanning started again, and the cycle was repeated eight times. We set the scanning speed to 50 mV/s.

The electrochemical impedance technique (EIS) can obtain electrode/electrolyte interface information and the kinetic information of electrode surface reactions. In the EIS test of the corrosion resistance of the coating in this paper, the frequency range was set as 99.0 kHz (100 kHz) 0.01 Hz, and the amplitude of the measured signal was selected as 5 mV. The sample needs to be tested for the open-circuit potential first and can be tested for impedance only when the surface condition is stable. In the kinetic comparative analysis

experiment of the deposition process of the coating, the four potential conditions relative to the reference electrode (−0.950, −1.050, −1.150, and −1.250 V) were tested, and the frequency range and amplitude of the test did not change. In this paper, the ZView software was used for fitting, and the data in ZView were used for analysis.

## 3. Results and Discussion

### 3.1. Effect of Sodium Molybdate Content on Cyclic Voltammetry Curve of Co-Mo Alloy Coating

The cyclic voltammetry curves detected in the electrolyte with different sodium molybdate concentrations are shown in Figure 1. As can be observed from the figure, when sodium molybdate was not added to the electrolyte, it can be observed from the curve variation trend that the current density remained at $0 \, \text{mA·cm}^{-2}$, with the negative potential moving to −0.758 V during the negative scanning process, which did not change, indicating that a Co reduction reaction had not occurred at this time. When the potential continued to shift negatively, the current gradually increased, and the wave current appeared, indicating that $Co^{2+}$ began the deposition process. A wide reduction peak occurs at −1.05 V, where the current density is about $-4.7 \, \text{mA·cm}^{-2}$, due to the reduction of $Co^{2+}$ to the metal Co. When the potential reaches −1.25 V, the positive scanning begins, and when the potential reaches −0.868 V, the current density changes to $0 \, \text{mA·cm}^{-2}$. A current loop was formed between −0.758 and −0.868 V, indicating that Co underwent nucleation on the electrode surface. Among these values, −0.758 V is the nucleation potential, which is represented by $E_{Nu}$, indicating that under this potential, Co begins to nucleate, and the reduction reaction occurs. In addition, −0.868 V is the equilibrium potential, represented by $E_{Co}$. The difference between the equilibrium potential and nucleation potential is the nucleation overpotential, denoted by NOP. When the positive scanning potential continued to drop to −0.4 V, a large oxidation peak appeared, and the peak current density reached $8.79 \, \text{mA·cm}^{-2}$. No oxidation peak appeared in the negative sweep process, indicating that the oxidation peak was caused by the dissolution of metal Co deposited on the electrode surface.

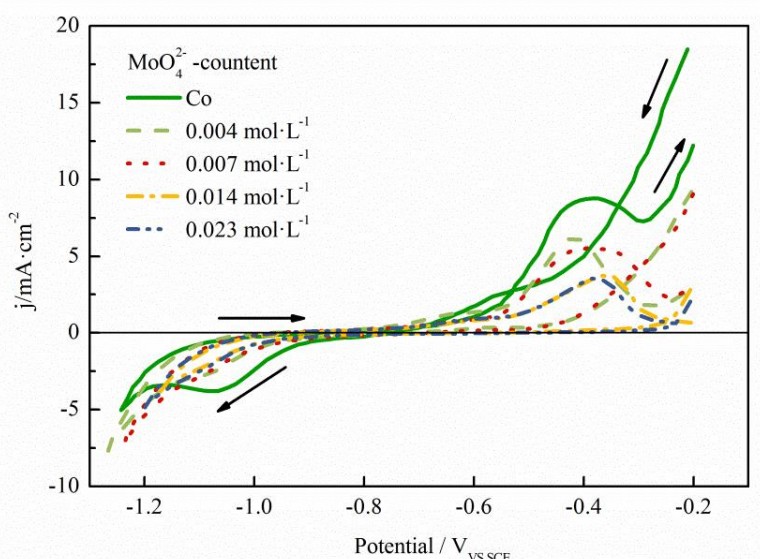

**Figure 1.** Cyclic voltammograms of Co and Co-Mo electrodeposition in different electrolytes.

When sodium molybdate was added to the electrolyte, it was found that the reduction peak characteristics at the −1.1 V position were weakened during the negative sweep. In addition, the reduction peak disappeared gradually with the increase in sodium molybdate content. It indicates that the Co-Mo co-deposition reaction is not very rapid at this time. By comparing the nucleation potential under different conditions in Table 2, it was found that with the increase in sodium molybdate content, the nucleation potential of the coating gradually became positive, that is, the trend of co-deposition became larger and larger,

the nucleation overpotential linearly increased, and the growth rate gradually decreased. The average grain size of deposited metal depends on the overpotential, and the average grain size of deposited metal is smaller at high overpotential. Combined with the working conditions of the plating solution, it can be considered that the reduction current at $-1.1$ V is low because the deposition process of Co is inhibited. The deposition potential of $MoO_4^{2-}$ is higher than that of $Co^{2+}$. However, because the pH of the plating bath is 9 and it is weakly alkaline, the deposition of $MoO_4^{2-}$ is inhibited to a certain extent, which weakens the reduction reaction current on the electrode. When the positive scanning has begun, it is found that the oxidation peak at $-0.4$ V decreases with the increase in sodium molybdate content, and the oxidation reaction on the electrode surface is not severe. It can be concluded that the introduction of the Mo element in the coating can inhibit the dissolution of the coating to a certain extent and improve the service life of the coating.

**Table 2.** The kinetic parameters for Co or Co-Mo electrodeposition in different electrolytes.

| $MoO_4^{2-}$ Content | $E_{Nu}$ | $E_{Co}$ | NOP |
|---|---|---|---|
| $mol \cdot L^{-1}$ | V | V | V |
| 0.000 | $-0.758$ | $-0.868$ | 0.110 |
| 0.004 | $-0.720$ | $-0.947$ | 0.227 |
| 0.007 | $-0.689$ | $-0.950$ | 0.261 |
| 0.014 | $-0.555$ | $-0.917$ | 0.362 |
| 0.023 | $-0.514$ | $-0.944$ | 0.430 |

Scanning linear voltammetry (LSV) was used for measurements to further understand the coating kinetics under different electrolyte conditions with different sodium molybdate contents. That is, the voltage between the test electrodes was controlled to linearly change, and the changes in the current with the voltage were recorded. Figure 2 shows the test results of electrolyte scanning linear voltammetry with different sodium molybdate contents. It can be found that when the potential conditions are the same, the current density value of the electrolyte-added sodium molybdate is higher, that is, the addition of sodium molybdate increases the degree of the cathode reduction reaction. This is because the deposition potential of $MoO_4^{2-}$ is higher than that of $Co^{2+}$, and because $MoO_4^{2-}$ is more likely to be deposited under the same conditions. It was also found that the reaction current density increased with the increase in sodium molybdate content when the potential remained unchanged. It indicates that the amount of sodium molybdate involved in the reaction increases and that the reaction becomes more violent. As can be observed from the polarization curve in Figure 2, the reaction current density of the coating increased significantly after the addition of sodium molybdate, which contradicted the phenomenon that the reduction peak of the electrolyte in the cathode region gradually decreased, as shown in Figure 1. This is because when cyclic voltammetry is used, the potential changes quickly and the steps become longer, so the error of the test results is larger. In addition, the lower limit of potential was set as high in the cyclic voltammetry test, and positive scanning began when it reached $-1.25$ V, at which time the reduction reaction current in the electrolyte was small. In a linear voltammetry test, the rate of potential change is slow, the lower limit is low, the reaction current gradually increases, and the metal elements in the electrolyte can react faster. With the increase in sodium molybdate content, the hydrogen evolution reaction of the coating was enhanced, and the polarization curve of the electrolyte fluctuated greatly. When the content of sodium molybdate in the plating solution reached 0.007 mol/L, the corrosion resistance of the coating was at its optimum (this research result was described in another paper on the microstructure and properties of Co-Mo coating). Combined with the research content in this chapter, the concentration of sodium molybdate in the plating solution was adjusted to 0.007 mol/L in the subsequent experiment.

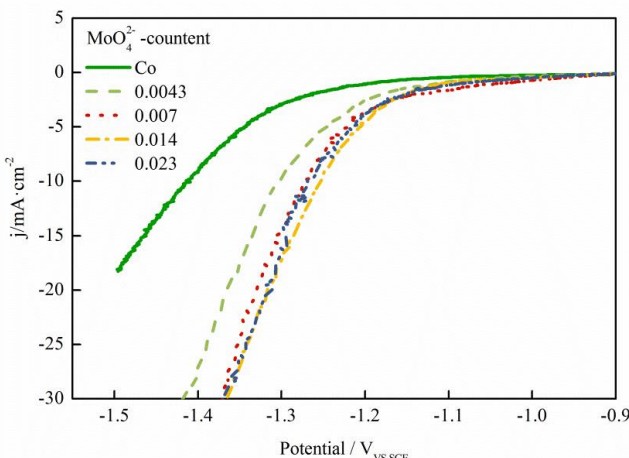

**Figure 2.** Linear sweep voltammograms of Co and Co-Mo electrodeposition in different electrolytes.

*3.2. Effect of Temperature on Cyclic Voltammetry Curve of Co-Mo Alloy Coating*

According to the study in the previous section, the optimal contents of sodium molybdate and $CoSO_4$ in the plating solution were set at 0.007 mol/L and 0.14 mol/L, respectively. This concentration was used in subsequent studies. Figure 3 shows the cyclic voltammetry test results of the electrolyte at different temperatures. Through observation, it can be found that when the electrolyte temperature is 30 °C, no obvious reduction peak characteristics can be observed in the negative scanning process. The bath used in this study uses cobalt sulfate as a cobalt source. In the sulfate system, a higher deposition temperature is usually required to obtain a coating with good properties and morphology. At low deposition temperatures, the coating deposition rate is slow, and the surface of the coating shows a uniform distribution of irregular patches. Especially in the range of 30–40 °C, the thickness of the deposited coating is very thin, the content of Co and Mo elements is also very low, and the deposition efficiency is poor. This may be because the ions in the bath move at a relatively slow rate at low temperatures. At 30 °C, fewer $Co^{2+}$ ions participated in the reduction reaction. With the increase in temperature, the characteristics of the reduction peak are gradually revealed. The acceleration of ion motion in the plating solution improves the deposition rate of the coating, the metal cations consumed by deposition are quickly replenished, the reduction reaction rate is improved, and the deposition process of the coating is promoted. At the same time, a higher deposition temperature is beneficial to weaken the concentration polarization effect of the cathode. This is also the main reason that the nucleation potential greatly fluctuates between 30 °C and 40 °C. When the positive scanning has begun, an obvious oxidation peak is observed at −0.4 V in the anode region, and the intensity of the oxidation peak gradually increases with the increase in electrolyte temperature. This phenomenon also occurs because the oxidation rate of the coating is affected by temperature. At the same time, according to the nucleation potential and equilibrium potential of the electrolyte at different temperatures shown in Table 3, it is found that with the increase in the electrolyte temperature, the plating nucleation overpotential gradually decreases; that is, the increase in temperatures is conducive to the plating nucleation process. However, a higher temperature will also cause adverse effects on the stability of the plating solution. Under the influence of internal stress, the stability of the plating solution, and the hydrogen evolution reaction, cracks appear on the surface of the coating and the corrosion resistance decreases. When the deposition temperature exceeds 60 °C, the reaction rate of hydrogen evolution on the surface of the coating accelerates when the temperature is too high, which has an impact on the deposition of the coating.

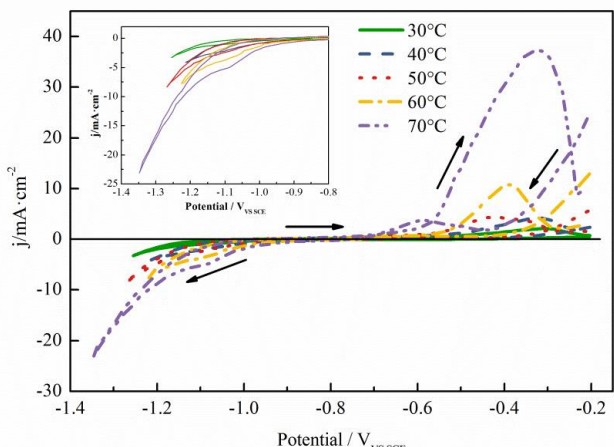

**Figure 3.** Cyclic voltammograms of Co-Mo electrodeposition in electrolytes with different temperatures.

**Table 3.** The kinetic parameters for Co-Mo electrodeposition in different temperatures.

| T/°C | $E_{Nu}$/V | $E_{Co}$/V | NOP/V |
|---|---|---|---|
| 30 | −0.498 | −0.931 | 0.433 |
| 40 | −0.723 | −0.959 | 0.236 |
| 50 | −0.711 | −0.973 | 0.262 |
| 60 | −0.749 | −0.948 | 0.199 |
| 70 | −0.748 | −0.946 | 0.198 |

Figure 4 shows the results of scanning the linear voltammetry of electrolytes at different temperatures. It can be observed from the figure that when the potential is constant, the reduction reaction rate increases and the current density gradually increases with the increase in electrolyte temperature. This is because the increase in temperature leads to the acceleration of the thermal motion of ions in the electrolyte, which increases the reaction current density. At the same time, a higher temperature can also reduce the polarization phenomenon of the electrode surface during the reaction process and also promote the reaction rate. In addition, by observing the fluctuation of linear polarization curves measured at different temperatures in the figure, it can be found that with the increase in electrolytic temperature, the detected polarization curve of the coating fluctuates because of hydrogen evolution. With the negative shift in potential, the effect of hydrogen evolution is enhanced, and the fluctuation of the polarization curve is increased.

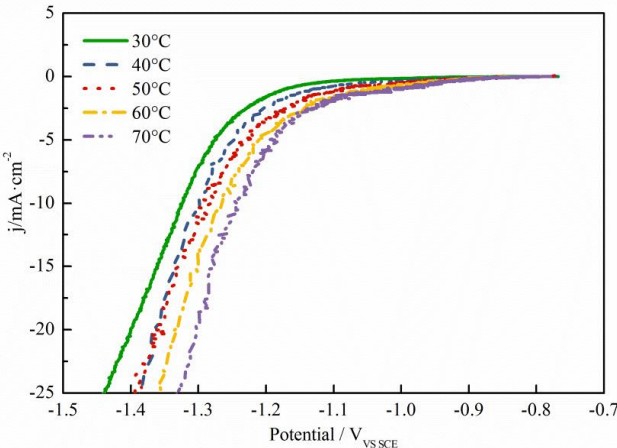

**Figure 4.** LSV of Co-Mo electrodeposition in electrolytes with different temperatures.

### 3.3. Effect of pH on Cyclic Voltammetry Curve of Co-Mo Alloy Coating

Figure 5 shows the cyclic voltammetry test results of electrodes under different pH electrolyte conditions. It can be found that the shape and position of the cyclic voltammetry curve significantly greatly with the change in electrolyte pH value. When the electrolyte pH is 7, no obvious reduction and oxidation peaks can be observed from the cyclic voltammetry curve. However, compared with the curve measured under other conditions, the current density of the electrode at this time is relatively large, indicating that the electrode surface is undergoing a REDOX reaction. This is because the closer the plating solution is to acidic conditions, the more favorable the reduction and deposition of the Mo element will be. When the pH value increased to 8, the current density in the electrode reaction process obviously decreased, and an obvious oxidation peak appeared at the anode region of −0.3 V. With the increase in electrolyte pH value, the reduction peak in the cathode zone gradually disappeared, and the reduction current density gradually decreased, indicating that the high pH value influenced the code position process of the coating. At the same time, it was found that when the electrolyte pH value was 9, the oxidation peak appeared at a position of −0.4 V in the anode region. When the pH value continued to increase, the oxidation peak in the anode region gradually decreased and moved to −0.2 V. Combined with the changes in the kinetic parameters in Table 4, it can be concluded that with the increase in electrolyte pH value, the inhibition effect of electrode reactions is enhanced, the equilibrium potential becomes negative, and the nucleation overpotential gradually increases.

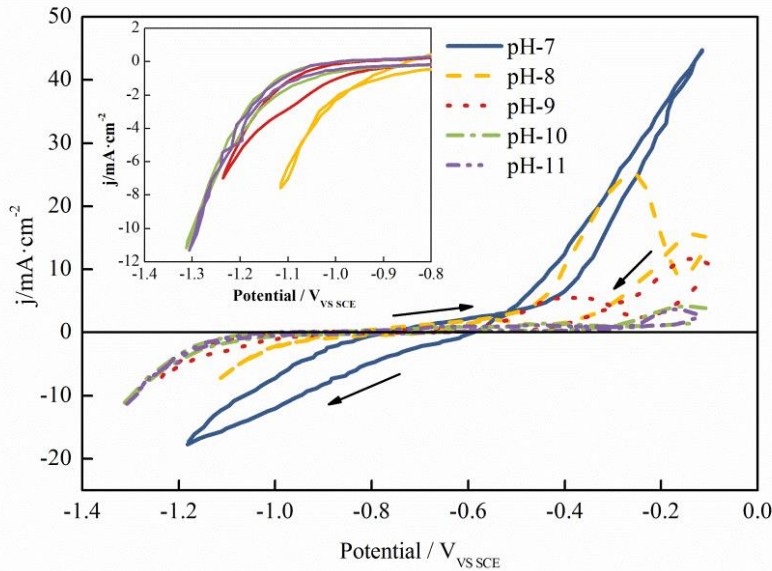

**Figure 5.** Cyclic voltammograms of Co-Mo electrodeposition in electrolytes with different pH values.

**Table 4.** The kinetic parameters for Co-Mo electrodeposition with different pH values.

| pH | $E_{Nu}$/V | $E_{Co}$/V | NOP/V |
|----|-----------|-----------|-------|
| 7 | −0.613 | −0.763 | 0.150 |
| 8 | −0.693 | −0.831 | 0.138 |
| 9 | −0.689 | −0.950 | 0.261 |
| 10 | −0.679 | −0.973 | 0.294 |
| 11 | −0.671 | −0.993 | 0.322 |

The linear voltammetry test results of electrodes under different pH electrolyte conditions are shown in Figure 6. With the increase in the pH value of the electrolyte, the reduction reaction rate on the electrode surface gradually decreases. When the potential is constant, the detected electrode reaction current density decreases gradually with the increase in electrolyte pH, and the reaction rate decreases. This indicates that the reduction reaction in the electrolyte is gradually strengthened by pH inhibition. Combined with the

trend of the polarization curve, it can be found that when the pH value of the electrolyte is between 10 and 11, the polarization curve almost coincides, indicating that under the condition of high pH, the electrode reduction process will not significantly change after being inhibited by the pH value to a certain extent. Combined with the deposition process of the coating, it can be concluded that this limit mainly depends on the composition of the plating bath and reaction temperature.

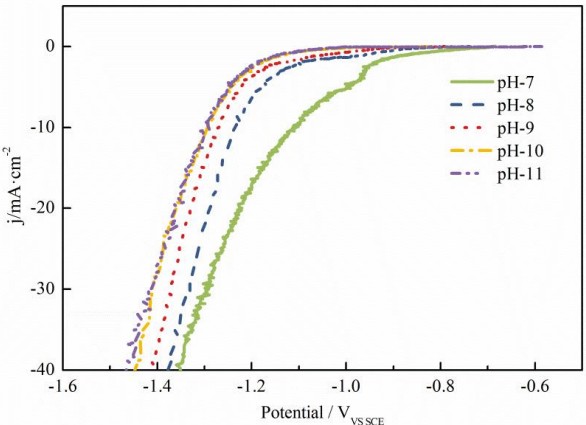

**Figure 6.** Linear sweep voltammograms of Co-Mo electrolytes with different pH values.

### 3.4. The Electrochemical Impedance Analysis of Co-Mo alloy Coating Electrode Surface

Figure 7 shows the linear voltammetry curve of the pure Co electrolyte and Co-Mo electrolyte when the concentration of sodium molybdate is 0.007 mol/L. Four groups of potential values of −0.950, −1.050, −1.150, and −1.250 V were selected for the electrochemical impedance test. By contrasting the impedance data of two electrolytes in various reaction states, the deposition kinetics of the Co-Mo coating was investigated (different potential values).

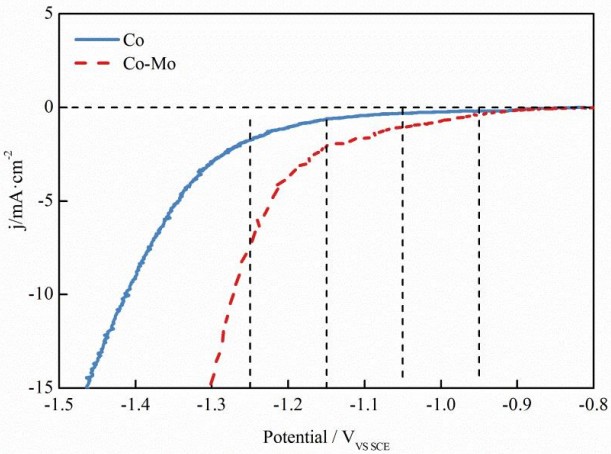

**Figure 7.** Linear sweep voltammograms of Co and Co-Mo electrolytes.

Figure 8 shows the electrochemical impedance test results of the electrolytes of Co and Co-Mo coatings at different deposition potentials. Through observation, it can be found that when the deposition potential is −0.950 V (Figure 8a), combined with the polarization curve of the electrolyte in Figure 7, the Co coating does not reach the deposition condition and the Co-Mo coating starts the deposition process. Following the electrochemical impedance spectroscopy at the current state, the Co electrolyte shows a characteristic of the time constant. The Co-Mo electrolyte presents two time-constant characteristics. The first time-constant characteristic of the Co-Mo coating electrolyte is a circular arc in the high- and middle-frequency regions, and the overall impedance value is large. This may be due to

the impedance generated by the joint action of the charge transfer on the electrode surface and the double-layer capacitance at the beginning of the electrode deposition. The second time-constant feature is a small arc in the low- and middle-frequency region, which may be generated by the diffusion movement of the reacting citrate complex $CoCit^-$, $[MoO_2\text{-}CoCit^-]_{ads}$ near the electrode surface. Currently, the reaction rate of the deposition process on the electrode surface is very low, so the deposition process is accompanied by adsorption and desorption phenomena.

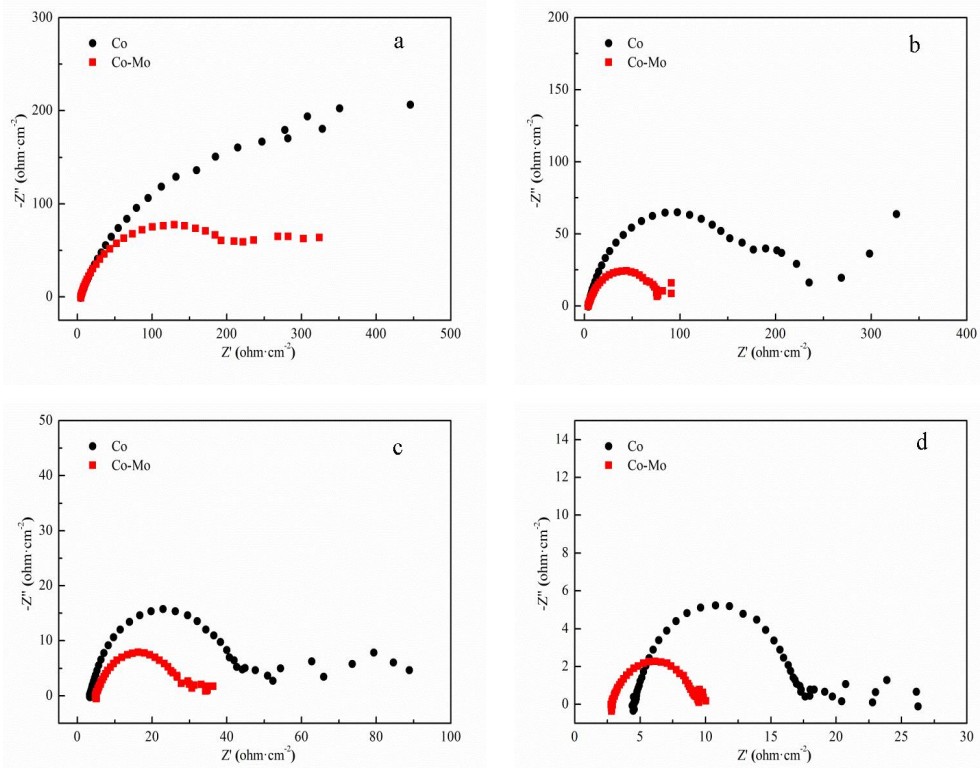

**Figure 8.** Nyquist plots of electroplating plating baths in different deposition potentials: ((**a**) −950 mV; (**b**) −1050 mV; (**c**) −1150 mV; (**d**) −1250 mV).

When the deposition potential is −1.050 V, the Co-Mo coating starts to deposit at a higher rate, while the Co coating begins the deposition process. Therefore, it can be observed that the electrolyte of the Co-Mo coating mainly presents two time-constant characteristics, and the impedance graph of the middle- and high-frequency region is an arc, representing the Faraday process of the electrode. The arc of the Faraday process decreases with the increase in deposition potential. At the same time, some data points from the diffusion process appear in the low-frequency region. The impedance diagram of the electrolyte of the Co coating shows three time constants. The first time constant is an arc located in the middle and high frequencies. The arc has a large impedance and represents the Faraday process of the electrode. The second time constant is a small arc in the low-frequency region. The formation reason is the same as the arc in the low-frequency region of Co-Mo at −0.950 V, which is generated by the adsorption and desorption phenomena of the citrate complex $CoCit^-$ near the electrode surface. The third time constant is a straight line inclined upward 45° in the low-frequency region, which indicates that there is evidence of diffusion phenomenon on the electrode surface, which is caused by the slow movement of ions in the electrolyte. When the deposition potential of the Co-Mo coating was −1.150 V, the coating showed two time-constant characteristics, corresponding to the Faraday process and the diffusion process, respectively, and the impedance value decreased gradually with the increase in deposition potential.

When the deposition potential exceeds −1.050 V, the Co coating begins to deposit, and three time-constant characteristics of the coating can be observed. The first time

constant is an arc in the middle-frequency region, representing the Faraday process of the electrode surface reaction. The second time constant is the arc in the low-frequency region, representing the double-layer capacitance and charge transfer process. The small line at the end of the low-frequency region represents the diffusion process, but because of the interference of the hydrogen evolution reaction, the distribution of the points in the line segment is chaotic. With the increase in deposition potential, the arc radius that represents the Faraday process of the coating decreases gradually, but the Co coating still presents a three-stage arc, and the analysis process is the same as before.

### 3.5. The Microstructure of the Coating under the Optimal Process Parameters

Figure 9 shows the surface micromorphology of a matrix Q235 substrate and pure Co and Co-Mo alloy coatings prepared under the same deposition conditions. Table 5 shows the optimum technology parameters for Co-Mo alloy coating preparation. Through observation, the surface of the substrate matrix material after grinding treatment (Figure 9a) has a parallel distribution of scratches, and there are also many holes. The pure Co coating obtained by deposition (Figure 9b) can completely cover the substrate surface and play a certain protective role. The Co-Mo coating prepared under the same conditions (Figure 9c) has good coverage on the matrix. Except for a few large cellular structures, the remaining cellular structures are almost the same size and evenly distributed on the surface of the matrix, without cracks, holes, and other defects, which can protect the matrix. The thickness of the prepared Co-Mo coating is about 10 μm (Figure 9d), and it has good bonding properties. There are no cracks or other defects between the substrate and the Co-Mo coating, which play a good isolation and protection role. The Co-Mo coating prepared in this paper under the optimal process is consistent with the data on the Co-Mo coating obtained by other scholars [19–22].

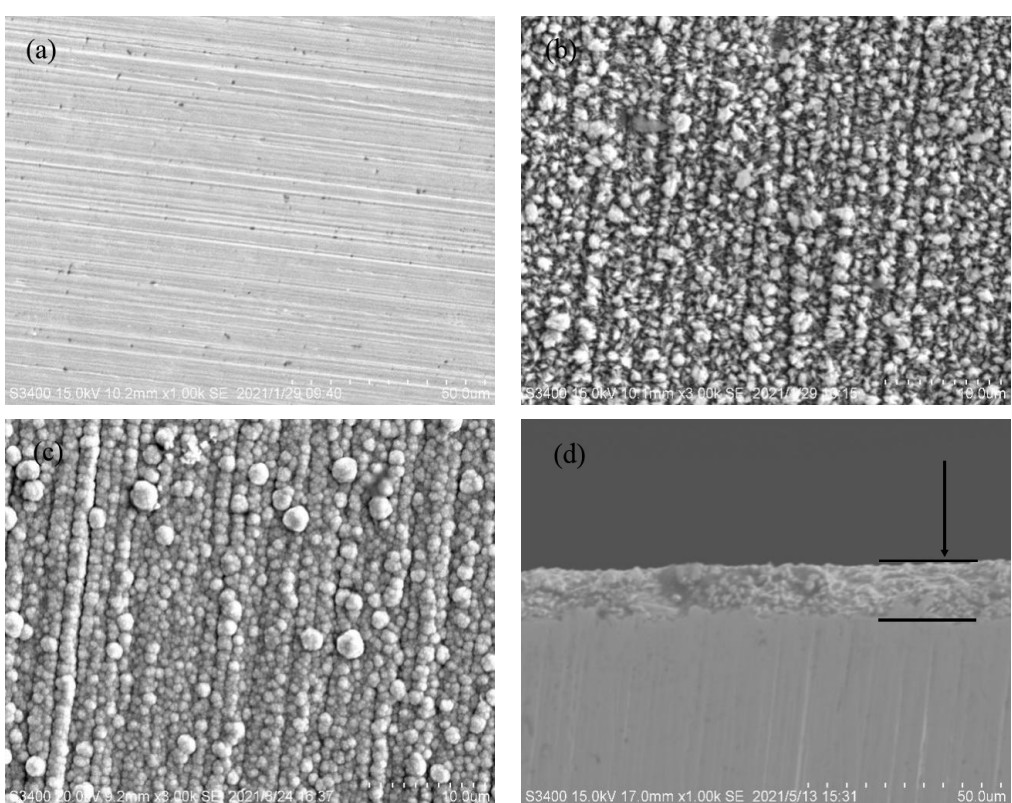

**Figure 9.** The SEM photograph of Q235, Co and Co-Mo alloy coatings: ((**a**) Q235; (**b**) Co coating; (**c**) Co-Mo alloy coating; (**d**) the thickness of Co-Mo alloy coating).

**Table 5.** Optimum technological parameters of Co-Mo alloy coating preparation.

| Category | Name | Condition |
|:---:|:---:|:---:|
| Plating bath composition | $CoSO_4 \cdot 7H_2O$ | 0.14 mol/L |
| | $Na_2MoO_4 \cdot 2H_2O$ | 0.007 mol/L |
| | $Na_3C_6H_5O_7 \cdot 2H_2O$ | 0.2 mol/L |
| Process parameters | Current density | 30 mA·cm$^{-2}$ |
| | pH | 9 |
| | Temperature | 50 °C |
| | Time | 3600 s |
| | Rotate speed | 280–300 rad/min |

Figures 10 and 11 show the element distribution and content of the coating prepared under the optimum process parameters. The coating is primarily composed of the four elements C, O, Co, and Mo. According to the solute type and working conditions in the coating, there are several sources of C and O elements. One of the sources is the plating bath itself. The other is sodium citrate from the plating bath. The uniform distribution of oxygen on the steel surface may be due to the significant enrichment of citrate in the coating. The rise in oxygen near the surface may also be due to the formation of native passive films that evenly cover the coating when it is exposed to air [27]. The citrate complex in the plating bath may be mixed and deposited on the coating during deposition. By observing the distribution of Co and Mo elements in the coating, it can be found that although the coating is distributed along the matrix scratch direction regarding its morphology, the distribution of elements in the coating is uniform, and there is no obvious agglomeration phenomenon. According to the EDS detection results, the main elements in the coating are Co (75.1 at.%) and Mo (19.17 at.%). The corrosion resistance of the coating is poor when the content of the Mo element is low. However, a high content of Mo will lead to hydrogen evolution, an increase in internal stress, surface pores, cracks, and other defects, which will also decrease the corrosion resistance of the coating. Combined with the test results, the optimal content of the Mo element in a Co-Mo alloy coating is 20 ± 1%, and the coating has the best corrosion resistance.

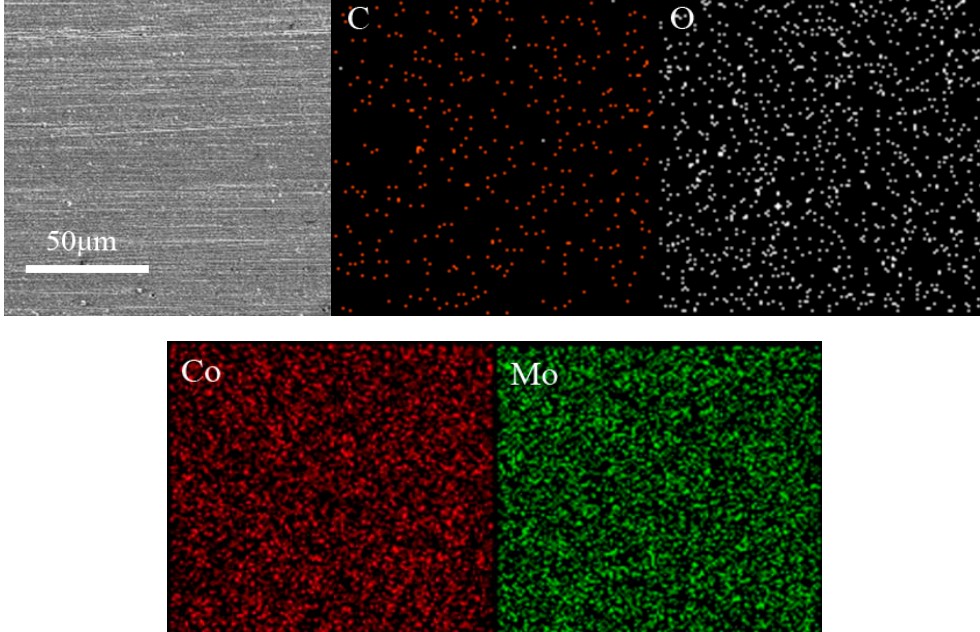

**Figure 10.** Element distribution on the surface of Co-Mo alloy coating.

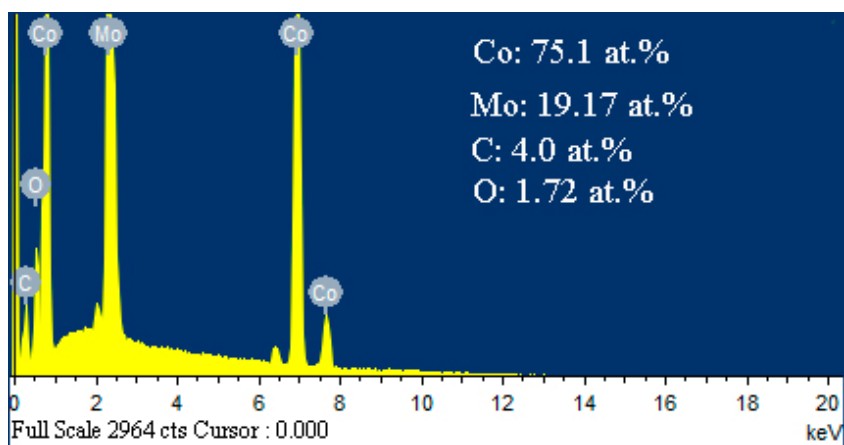

**Figure 11.** EDS spectrum and chemical composition of Co-Mo alloy coating.

## 4. Conclusions

In this paper, the kinetic process of electrodeposition of a Co-Mo coating was studied using cyclic voltammetry and electrochemical impedance. The effects of sodium molybdate content, deposition temperature, pH value, and other technological parameters on the deposition kinetics of alloy coatings were studied. The deposition processes of Co and Co-Mo coatings at different potentials were studied by electrochemical impedance testing. The results are as follows:

(1)  As the content of sodium molybdate increases, more Mo elements participate in the deposition process, and the current density of the cathode reduction reaction increases. At the same time, a higher Mo content can reduce the dissolution of the coating as an anode and improve the service life of the coating. However, excessive Mo element content will lead to cracks on the surface of the coating and affect its performance.

(2)  The deposition temperature will affect the processes of reduction deposition and oxidation dissolution of the coating. At low deposition temperatures, the deposition rate of the coating is slow, especially in the range of 30–40 °C, the deposited coating thickness is very thin, the content of Co and Mo elements is also very low, and the deposition efficiency is poor. This may be because the ions in the bath move at a relatively slow rate at low temperatures. With the increase in temperature, the characteristics of the reduction peak gradually become clearer, and the speed of ion movement in the bath increases the deposition rate of the coating, which promotes the deposition process of the coating. However, higher temperatures will also cause adverse effects on the stability of the plating solution. A high-quality coating with excellent corrosion resistance can be obtained at 50 °C with high deposition efficiency.

(3)  The pH of the plating bath will directly affect the reduction reaction process of $MoO_4^{2-}$ ions in the plating bath. With the increase in pH value, the reduction reaction rate of $MoO_4^{2-}$ ions decreases, and the cathodic reduction reaction current density decreases. At the same time, the peak current density of anodizing decreases with the decrease in the pH value of the plating solution, indicating that the alkaline plating solution has an inhibitory effect on the plating dissolution process.

(4)  In the deposition process, the Co coating maintains three time-constant characteristics, which represent the Faraday process, the double-layer capacitance and charge transfer process, and the diffusion process. The Co-Mo coating maintains two time-constant characteristics during deposition, corresponding to the Faraday process and diffusion process, and the Faraday process is the main process.

(5)  Based on the studies performed, the optimal coating parameters are as follows (such as those shown in Table 5): the main bath composition was adjusted to $CoSO_4 \cdot 7H_2O$ (0.14 mol/L), $Na_2MoO_4 \cdot 2H_2O$ (0.007 mol/L) and $Na_3C_6H_5O_7 \cdot 2H_2O$ (0.2 mol/L). We adjusted the plating solution pH (9) and used a constant temperature magnetic agitator

to stir the composite plating solution (280–300 rad/min) before and during plating. We selected a current density of 30 mA·cm$^{-2}$ and a temperature of 50 °C. Under the above technological parameters, the Co-Mo coating prepared by electrodeposition at 3600 s has better performance.

**Author Contributions:** Investigation, S.X.; Resources, X.Z. and Q.Z.; Writing—original draft, F.W.; Writing—review and editing, X.N. All authors have read and agreed to the published version of the manuscript.

**Funding:** This publication was financed by the 2022 Emerging Interdisciplinary Construction Project (grant numbers Z1010122003) and Shaanxi Province Agricultural Mechanization Special Project (grant numbers Z20221019).

**Institutional Review Board Statement:** Not applicable.

**Informed Consent Statement:** Not applicable.

**Data Availability Statement:** Not applicable.

**Acknowledgments:** We would like to acknowledge the College of Mechanical and Electronic Engineering of Northwest A&F University for their full support of this study, and to Qiongyu Zhou of Foshan University for his guidance.

**Conflicts of Interest:** The authors declare no conflict of interest.

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
