# Peer review of "Effect of Process Parameters on Electrodeposition Process of Co-Mo Alloy Coatings"

_coatings, doi:10.3390/coatings13040665_

Round 1
Reviewer 1 Report
Reviewer Recommendation and Comments for manuscript coatings-2293402 with the title: “Effect of process parameters on electrodeposition process of Co-Mo alloy coatings”, authors: X. Nan, F. Wang, S. Xin, X. Zhu, Q. Zhou.
The authors present the electrodeposition of Co-Mo coating on Q235 steel. Electrochemical deposition parameters such as sodium molybdate content, deposition temperature and pH value are studied.
The main comments that I find useful for improving the quality of the article are presented below:
*│The manuscript must be proofread by a native English speaker.
*│Authors' names and affiliations should be verified. There is no affiliation c!
*line 61│”H.Krawiec” the typo must be corrected
*line 79│”(0.14mol/L) (0.0047-0.0233mol/L) (0.2mol/L).” the typo must be corrected
*line 81│”two-electrode system” (WE, CE and RE) three-electrode system
*line 84│”PH” the typo must be corrected
*│According to ASTM, Q235 steel is of four grades (Q235A, Q235B, Q235C and Q235D) and none of them contain Ni. Also, the Mn content is much higher. What type of steel was used (according to ASTM)?
*line 121│”The reaction current density of the electrode surface during the test was not more than 1 mA/cm2.” What does the reaction current density of the electrode surface mean?
*line 125│”change characteristics of electrode reaction process”. The manuscript must be proofread by a native English speaker.
*line 131│”50mV/s.” the typo must be corrected
*line 135│”99.0kHz (100kHz)~0.01Hz 5mV.” the typo must be corrected
*│Figure 1 should also contain the cyclic voltammogram corresponding to the supporting electrolyte (citrate only).
*│Figure 1 must also contain the cyclic voltammogram corresponding to the electrodeposition of Mo (citrate+molybdate)
*line 170-173│The potentials (ENU) shown in Table 2 cannot be observed in Figure 1!
*line 191│”the addition of sodium molybdate increases the degree of cathode reduction reaction”. Figure 2 shows that at the highest concentrations (0.007, 0.014 and 0.023) no difference can be made. Processes are influenced to a lesser extent by changes in concentration.
*Fig. 3 / Fig. 4│The concentrations of the ionic species (Co2+ and MoO42-) in the electrolyte solutions are not specified.
*line 217│”the reduction reaction rate is gradually accelerated”. Acceleration means something else. In this context, the term acceleration is used inappropriately.
*line 234│”reaction rate accelerates”. Acceleration means something else. In this context, the term acceleration is used inappropriately.
*│Graphics quality needs to be improved. The background of the figures is gray!?
*Fig. 5 / Fig. 6│The concentrations of the ionic species (Co2+ and MoO42-) in the electrolyte solutions are not specified.
*table 5│”PH” the typo must be corrected
*│The coating also contains oxygen. Is it possible that the coating contains molybdenum oxides?
*│It could be a confusion between " Acknowledgments " and "Funding". Must be checked.
*The Coatings journal require a specific format of references, authors must pay more attention in their writing. No reference is written according to the format required by the journal.
*There are some grammar and typing mistakes.
*The authors must revise the entire manuscript.
Author Response
Comment 1: *│The manuscript must be proofread by a native English speaker.
Response: We regret there were problems with the English. The paper has been carefully revised by a native English speaker to improve the grammar and readability.
Comment 2: *│Authors' names and affiliations should be verified. There is no affiliation c!
Response: I'm sorry that our lack of rigor has led to this problem. We have revised it in the manuscript.
Comment 3: *line 61│”H.Krawiec” the typo must be corrected.
Response: I'm sorry that our lack of rigor has led to this problem. We have revised it in the manuscript.
Comment 4: *line 79│”(0.14mol/L) (0.0047-0.0233mol/L) (0.2mol/L).” the typo must be corrected.
Response: I'm sorry that our lack of rigor has led to this problem. We have revised it in the manuscript.
Comment 5: *line 81│”two-electrode system” (WE, CE and RE) three-electrode system.
Response: We thank the reviewer for raising this question. We have revised it in the manuscript.
Comment 6: *line 84│”PH” the typo must be corrected.
Response: I'm sorry that our lack of rigor has led to this problem. We have revised it in the manuscript.
Comment 7: *│According to ASTM, Q235 steel is of four grades (Q235A, Q235B, Q235C and Q235D) and none of them contain Ni. Also, the Mn content is much higher. What type of steel was used (according to ASTM)?
Response: I am very sorry for this problem, which is a major mistake of the author, because in another article, a series of studies were carried out with manganese steel as the matrix, and the data of 65 manganese steel was mistakenly copied over when writing. In this study, standard Q235B steel was used. Sorry again, we have corrected it in the manuscript.
Comment 8: *line 121│”The reaction current density of the electrode surface during the test was not more than 1 mA/cm2.” What does the reaction current density of the electrode surface mean?
Response: We are so grateful for your kind question. Here the reaction current density at the electrode surface is a set value of the instrument during the measurement process. The voltage is gradually increased, so is the current on the electrode surface, so it is necessary to add an upper limit, because it was not described clearly when writing the manuscript, and now descriptive sentences have been added.
Comment 9: *line 125│”change characteristics of electrode reaction process”. The manuscript must be proofread by a native English speaker.
Response: We regret there were problems with the English. The paper has been carefully revised by [a native English speaker]/[a professional language editing service] to improve the grammar and readability.
Comment 10: *line 131│”50mV/s.” the typo must be corrected.
Response: I'm sorry that our lack of rigor has led to this problem. We have revised it in the manuscript.
Comment 11: *line 135│”99.0kHz (100kHz)~0.01Hz 5mV.” the typo must be corrected.
Response: I'm sorry that our lack of rigor has led to this problem. We have revised it in the manuscript.
Comment 12: *│Figure 1 should also contain the cyclic voltammogram corresponding to the supporting electrolyte (citrate only).
Response: We thank the reviewer for raising this question. Sodium citrate, as a complexing agent, plays an important role in the plating solution, reducing the loss of Co2+ while inducing MoO42- deposition. However, this paper mainly focused on the influence of sodium molybdate as the main salt on the deposition process, and the influence of sodium citrate on Co-Mo coating has been studied in the paper of Elvira Gómez (Gómez, E.; Pellicer, E.; Vallés, E. Influence of the bath composition and the pH on the induced cobalt–molybdenum electrodeposition. Journal of Electroanalytical Chemistry 2003, 556, 137-145, doi:10.1016/s0022-0728(03)00339-5.). To avoid the repetition of his research content, sodium citrate was not studied.
Comment 13: *│Figure 1 must also contain the cyclic voltammogram corresponding to the electrodeposition of Mo (citrate + molybdate).
Response: We thank the reviewer for raising this question. The reason for this problem is like the response in comment 12, since Mo alone cannot be obtained by electrodeposition, it needs to be deposited by Co coordination, with sodium citrate used as the complexing agent. The emphasis is on studying the effect of MoO42- on deposition, and to avoid the duplication of research.
Comment 14: *line 170-173│The potentials (ENU) shown in Table 2 cannot be observed in Figure 1!
Response: We thank the reviewer for raising this question. In the electrodeposition process, the content of molybdate ion is not high, and the elevation brought by the shape nuclear site is not obvious, which is not easy to observe. This phenomenon was studied in part of Dr. Zhou Qiongyu's doctoral dissertation, in which the change of nuclear potential was observed and the mechanism of electrocrystallization was studied by using timing ampere-curve and other methods. So, it is a common phenomenon that the nuclear potential in Figure 1 is not easily observed.
Comment 15: *line 191│”the addition of sodium molybdate increases the degree of cathode reduction reaction”. Figure 2 shows that at the highest concentrations (0.007, 0.014 and 0.023) no difference can be made. Processes are influenced to a lesser extent by changes in concentration.
Response: We thank the reviewer for raising this question. Co was used to induce sodium molybdate deposition on the substrate surface. When the content of Co is certain and the concentration of sodium molybdate reaches a certain level, the deposition will not continue, that is, the deposition of Mo has reached its peak. Therefore, when the concentration reaches 0.007, the process is less affected by the concentration change.
Comment 16: *Fig. 3 / Fig. 4│The concentrations of the ionic species (Co2+ and MoO42-) in the electrolyte solutions are not specified.
Response: I'm sorry that our lack of rigor has led to this problem. We have added corresponding descriptions to the end of 3.1 and the beginning of 3.2 in the manuscript.
Comment 17: *line 217│”the reduction reaction rate is gradually accelerated”. Acceleration means something else. In this context, the term acceleration is used inappropriately.
Response: We regret there were problems with the English. We have replaced accelerated with increased.
Comment 18: *line 234│”reaction rate accelerates”. Acceleration means something else. In this context, the term acceleration is used inappropriately.
Response: We regret there were problems with the English. We have replaced accelerated with increases.
Comment 19: *│Graphics quality needs to be improved. The background of the figures is gray!?
Response: We thank the reviewer for raising this question. Some graphics have been reworked.
Comment 20: *Fig. 5 / Fig. 6│The concentrations of the ionic species (Co2+ and MoO42-) in the electrolyte solutions are not specified.
Response: I'm sorry that our lack of rigor has led to this problem. The reason for this problem is like the response in comment 15. We have added corresponding descriptions to the end of 3.1 and the beginning of 3.2 in the manuscript.
Comment 21: *table 5│”PH” the typo must be corrected.
Response: I'm sorry that our lack of rigor has led to this problem. We have revised it in the manuscript.
Comment 22: *│The coating also contains oxygen. Is it possible that the coating contains molybdenum oxides?
Response: We thank the reviewer for raising this question. No molybdenum oxide was observed in the XRD test of the coating. In the deposition process, Co2+ is reduced to Co by reaction, while MoO42- obtains electrons and is reduced to molybdenum oxide, which is further reduced to Mo under the action of citrate complex, and finally forms Co-Mo alloy. Molybdenum oxide is reduced as an intermediate product.
Comment 23: *│It could be a confusion between " Acknowledgments " and "Funding". Must be checked.
Response: We thank the reviewer for raising this question. We have revised it in the manuscript.
Comment 24: *The Coatings journal require a specific format of references, authors must pay more attention in their writing. No reference is written according to the format required by the journal.
Response: I'm sorry that our lack of rigor has led to this problem. We have revised it in the manuscript.
Comment 25: *There are some grammars and typing mistakes.
Response: We regret there were problems with the English. The paper has been carefully revised by a native English speaker.
Comment 26: *The authors must revise the entire manuscript.
Response: Thank you very much for the reviewer's comments on this article. The article has been modified according to the reviewer's comments.
Reviewer 2 Report
The authors have worked hard and presented a good work. Thank you.
the authors can make figure 11 more visible and vertical measurement for a better comparison of this manuscript. In addition, the Results and Discussion section can be expanded a little more.
Author Response
Comment 1: The authors can make figure 11 more visible and vertical measurement for a better comparison of this manuscript. In addition, the Results and Discussion section can be expanded a little more.
Response: Thank you very much for the reviewer's advice. We have reworked Fig.11 and added element content to it to make it easier to understand and clearer. At the same time, the content of optimal process parameters is also supplemented for the results and discussion.
Reviewer 3 Report
The study is quite interesting, but there are a number of questions and comments:
1) Section "Introduction". Didn't the kinetics of the process be studied in [19-22]? Emphasize the scientific novelty of your work.
2) Section "Materials and Methods". It is necessary to clarify why Na3C6H5O7 is used as an additive to the electrolytic deposition solution. Why exactly in the specified concentration?
3) Section "Results". Table 2. With an increase in the concentration of molybdenum in the solution, the NOP index increases. Have you tried plotting dependencies? Is growth linear or not?
4) Section "Results". Fig. 3 and Table 3. For what concentration of molybdenum ions are the data given? Why does the nucleation potential of cobalt change so sharply when going from 30°C to 40°C?
5) Fig. 11. Give numerical data in wt. % by the ratio of elements in the resulting coating, and preferably by the thickness of the coating. Will the elements be evenly distributed?
6) Do your data on the optimal mode of coating coincide with the data of works [19-22]? An appropriate discussion should be added to the text of the manuscript.
7) Section "Conclusion". Indicate that, based on the studies performed, the optimal coating parameters are as follows (and provide these data).
Author Response
Comment 1: Section "Introduction". Didn't the kinetics of the process be studied in [19-22]? Emphasize the scientific novelty of your work.
Response: We thank the reviewer for raising this question. In these studies, the study of process parameters is only a simple description, not involved in electrodeposition kinetics. A description has been added to the appropriate section of the introduction.
Comment 2: Section "Materials and Methods". It is necessary to clarify why Na3C6H5O7 is used as an additive to the electrolytic deposition solution. Why exactly in the specified concentration?
Response: We thank the reviewer for raising this question. According to Elvira Gómez et al.( Gómez, E.; Pellicer, E.; Vallés, E. Influence of the bath composition and the pH on the induced cobalt–molybdenum electrodeposition. Journal of Electroanalytical Chemistry 2003, 556, 137-145, doi:10.1016/s0022-0728(03)00339-5.), in alkaline plating solution, Na3C6H5O is the complexing agent and plays an important role in plating solution. The addition of Na3C6H5O can make the Co2+ in the coating exist in the form of citrate complex CoCit- and reduce the Co2+ loss. At the same time CoCit- is also an important part of MoO42- induced to change into metal Mo during the deposition process. In addition, sodium citrate and H+ dissociation equilibrium, can play a buffer effect. According to their result, 0.2 mol citrate assured coherent homogeneous Co-Mo deposits. A description has been added to the appropriate section of the introduction.
Comment 3: Section "Results". Table 2. With an increase in the concentration of molybdenum in the solution, the NOP index increases. Have you tried plotting dependencies? Is growth linear or not?
Response: We thank the reviewer for raising this question. With the increase of molybdate concentration, the nucleation overpotential NOP increased linearly and the growth rate decreased gradually. The average grain size of deposited metal depends on overpotential, and the average grain size of deposited metal is smaller at high overpotential. We have added part of the description of nucleation overpotential.
Comment 4: Section "Results". Fig. 3 and Table 3. For what concentration of molybdenum ions are the data given? Why does the nucleation potential of cobalt change so sharply when going from 30°C to 40°C?
Response: We thank the reviewer for raising this question. In the study of this paper, the influence of temperature on the deposition rate is still great. The increase of bath temperature is conducive to the diffusion and migration of metal ions in the bath. In the experiment, the content of Mo element is low when the coating is at 30°, and the deposition rate accelerates with the increase of Mo element content after 40°. In the range of 30° to 40°, the thermal motion of ions in the electrolyte gradually becomes obvious.
Comment 5: Fig. 11. Give numerical data in wt. % by the ratio of elements in the resulting coating, and preferably by the thickness of the coating. Will the elements be evenly distributed?
Response: I'm sorry that our lack of rigor has led to this problem. The unit of element content in Figure 11 and the corresponding description should be at.%, which is not made clear in the paper due to the author's negligence. We have modified this description and reworked Figure 11.
Comment 6: Do your data on the optimal mode of coating coincide with the data of works [19-22]? An appropriate discussion should be added to the text of the manuscript.
Response: We thank the reviewer for raising this question. Co-Mo coating prepared in this paper under the optimal process are basically consistent with the data of the Co-Mo coating obtained by other scholars [19-22], but the properties of the Co-Mo coating are worse than that of the composite coating with nanoparticles added[21-22]. In another part of our work, we have studied the addition of nanoparticles, which is not involved in this paper. The corresponding description has been added in Article 3.5.
Comment 7: Section "Conclusion". Indicate that, based on the studies performed, the optimal coating parameters are as follows (and provide these data).
Response: Thank you very much for the reviewer's comments on this article. The article has been modified according to the reviewer's comments.
Round 2
Reviewer 1 Report
Reviewer Recommendation and Comments for manuscript coatings-2293402 with the title: “Effect of process parameters on electrodeposition process of Co-Mo alloy coatings”, authors: X. Nan, F. Wang, S. Xin, X. Zhu, Q. Zhou.
The authors consider the referees' comments and provide a new, much improved version of the manuscript. However, I think the authors need to check and proofread the manuscript much more carefully. From a scientific point of view, I have only one unclear point, namely:
Comment 22: *│The coating also contains oxygen. Is it possible that the coating contains molybdenum oxides?
Response: We thank the reviewer for raising this question. No molybdenum oxide was observed in the XRD test of the coating. In the deposition process, Co2+ is reduced to Co by reaction, while MoO42- obtains electrons and is reduced to molybdenum oxide, which is further reduced to Mo under the action of citrate complex, and finally forms Co-Mo alloy. Molybdenum oxide is reduced as an intermediate product.
Comment: The EDS analysis (Figure 11) indicates 1.72% oxygen, while the elemental distribution map (Figure 10) indicates a uniform distribution of oxygen on the steel surface. How is the presence of 1.72% oxygen in the coating explained?
There are still many mistakes in English and editing.
*page 13 is blank
*line 89│”0.2 mol citrate” means mole L-1? mol is not concentration, and without specifying volume, the molar quantity cannot express concentration.
*line124│“exposed to 3.5wt%NaCl solution”
*line 214│”the kinetic” / The kinetic
etc.
etc.
*”Author contribution” is missing.
*The Coatings journal require a specific format of references, authors must pay more attention in their writing. No reference is written according to the format required by the journal.
Reviewer 3 Report
It seems to me that it is worth adding a discussion of the progress of further research, especially on the effect of temperature on the nucleation potential. From the data obtained in Table 3 it is clearly seen that in the range from 30 to 40 °Ð¡ there are curious physicochemical phenomena that change the numerical value of the potential so sharply.
Author Response
Comment 1: It seems to me that it is worth adding a discussion of the progress of further research, especially on the effect of temperature on the nucleation potential. From the data obtained in Table 3, it is clearly seen that in the range from 30 to 40 °Ð¡, there are curious physicochemical phenomena that change the numerical value of the potential so sharply.
Response: Thank the reviewer for your valuable comments. According to your suggestion, we further discussed the research content in Section 3.2, and focused on analyzing the reasons for the large fluctuation of the nucleation potential between 30℃ and 40℃. And we rewrote the conclusion.